# Breast Cancer-Related Lymphedema: Recent Updates on Diagnosis, Severity and Available Treatments

**DOI:** 10.3390/jpm11050402

**Published:** 2021-05-12

**Authors:** Marco Pappalardo, Marta Starnoni, Gianluca Franceschini, Alessio Baccarani, Giorgio De Santis

**Affiliations:** 1Division of Plastic and Reconstructive Surgery, Department of Medical and Surgical Sciences, Modena Policlinico Hospital, University of Modena and Reggio Emilia, 41124 Modena, Italy; marco.pappalardo@unimore.it (M.P.); alessio.baccarani@unimore.it (A.B.); giorgio.desantis@unimore.it (G.D.S.); 2Clinical and Experimental Medicine PhD Program, University of Modena and Reggio Emilia, 41124 Modena, Italy; 3Multidisciplinary Breast Center, Department of Woman and Child Health and Public Health, Fondazione Policlinico Universitario Agostino Gemelli IRCCS, Università Cattolica del Sacro Cuore, Largo A. Gemelli, 8-00168 Rome, Italy; gianlucafranceschini70@gmail.com

**Keywords:** breast cancer, lymphedema, lymphaticovenous anastomosis, vascularized lymph node transfer, lymphatic microsurgery, radiotherapy

## Abstract

Breast cancer-related lymphedema (BCRL) represents a global healthcare issue affecting the emotional and life quality of breast cancer survivors significantly. The clinical presentation is characterized by swelling of the affected upper limb, that may be accompanied by atrophic skin findings, pain and recurrent cellulitis. Cardinal principles of lymphedema management are the use of complex decongestive therapy and patient education. Recently, new microsurgery procedures have been reported with interesting results, bringing in a new opportunity to care postmastectomy lymphedema. However, many aspects of the disease are still debated in the medical community, including clinical examination, imaging techniques, patient selection and proper treatment. Here we will review these aspects and the current literature.

## 1. Introduction

Breast cancer-related lymphoedema (BCRL) remains a significant clinical issue for breast cancer survivors in that it causes severe physical and psychological discomfort. With the ever-increasing incidence of breast cancer, more patients are undergoing breast surgery that may include sentinel lymph node biopsy (SLNB) and/or axillary lymph node dissection (ALND) [1,2]. Chest wall radiotherapy is also commonly performed in patients with previous ALND, whereas axillary radiotherapy is sometimes indicated as an alternative to ALND in selected patients [3,4]. Both axillary surgery or radiotherapy can cause lymphedema with significant impairment of the normal lymphatic drainage producing an abnormal collection of protein-rich fluid within the upper limb. Despite improved early detection and evolving approaches to minimize surgical intervention increasing conservative surgery procedures with fewer ALND [5]; BCRL remains however a significant healthcare burden [6].

According to reports the incidence of BCRL varies and is approximately 20% at one year and increases to 40% at ten years after breast cancer treatment with a cumulative incidence of 28% [4,7]. Indeed, lymphedema is significantly more likely to occur following ALND than after SLNB alone [8,9]. Lymphedema can to develop within days postoperatively and can continue to present until 11 years after breast cancer treatment [10].

The impact of a lower quality-of-life on patients with lymphedema is unquestionable and there is a higher likelihood of poorer general health [11]. Besides, complications of lymphedema including repeated episodes of cellulitis and ulceration, may require antibiotic therapy and hospitalization.

Cardinal principles of lymphedema treatment are patient education and control of concomitant diseases that may worsen swelling. Upper limb swelling is primarily controlled through the use of complex decongestive therapy (CDT) such as manual lymphatic drainage, bandages, compression garments and individualized exercises to reduce limb swelling [12]. Historical surgical treatments for lymphedema such as Homans’ operation and Charles’ procedure are palliative and nowadays largely abandoned [13]. Instead, a more recent volume reduction approach is circumferential liposuction [14,15]. In recent years, microsurgical and supermicrosurgical techniques, such as lymphaticovenous anastomosis (LVA) [16,17] and vascularized lymph node transfer (VLNT) [18] have gained popularity as they can potentially reconstitute lymphatic flow and, ideally, reduce the use of compression garments.

The recent introduction of severity staging using lymphoscintigraphy [19,20], and indocyanine green (ICG) [16,21] has helped the patient selection and improved the reported outcomes as it allows preoperatively to evaluate the lymphatic obstruction and the lymphatic flow patterns. This review article will focus on the current issues and debates in BCRL including diagnosis, severity, patient selection criteria and type of treatment available.

## 2. Diagnosis of BCRL and Clinical Symptoms

In order to properly manage upper limb lymphedema, the physician should first have a detailed knowledge of the diagnosis and severity of the disease. Traditionally health-care professionals have clinically diagnosed BCRL with subjective interpretations of swelling [22]. Diagnosis of upper limb lymphedema depends on a combination of comprehensive history, physical examination with subjective/objective symptoms and physiologic measures [6]. The patient’s medical history including risk evaluation, medical conditions and medications that may cause edema should be meticulously reviewed. The differential diagnosis of BCRL is wide and can include: infection, congestive heart failure, primary/recurrent malignancy, vascular anomalies, electrolyte imbalances, hypo-proteinemia, renal or hepatic failure, and peripheral neuropathies [23]. The common subjective clinical symptoms of patients with lymphedema in the upper limb are swelling, numbness, heaviness, tightness, stiffness, decreased coordination and mobility, limb fatigue or weakness. However, symptom presentation is broad and not all patients experience these symptoms. Next, during the physical examination, evaluation of the swollen limb should provide information regarding size, presence of scars, comparison with the healthy limb, skin condition and sensation. Objective clinical signs can include skin changes such as reddening, hyperkeratosis, thickening/firmness of tissues. Pitting edema is commonly seen at the end of the latent phase, with a depression formed in the skin after a fingertip pressure as the lymph is pushed into the surroundings. Later, non-pitting edema is characterized by hypertrophied adipose tissue with fibrosis. Stemmer’s test is commonly performed and it is considered positive when it is difficult or impossible to pinch the skin at the base of the toes or at proximal phalanx of the fingers due to severe fibrosis. Patients with BCRL are susceptible to recurrent episodes of cellulitis that may increase adipose tissue deposition [24].

Limb volumetric measurements are considered the mainstay of the diagnosis and to track the progression of the disease. Many non-invasive tools such as tape circumferential measurements, water displacement, perometry, bioimpedence spectroscopy and three-dimensional laser scanning are available to measure lymphedema (Table 1). However, there is not a universally accepted method.

### 2.1. Tape Circumferential Measurements

Circumferential limb measurements at designated anatomic distances are the most common and easy method for quantification of lymphedema by measuring limb size or girth. Generally, a circumferential difference of greater than 2 cm or a volumetric differential of more than 200 mL is considered significant [25]. Sequential circumference measurements measured at standardized anatomical locations are widely used. The distance of each designated point is measured and total upper limb volume calculated based on the truncated cone formula [26].

Cheng et al. have described a sequence of measurements at 10 cm proximal and distal to the elbow [27,28]. These data are compared to the healthy limb, producing a quantitative limb measurement of lymphedema as well as a tool to check the progress during the follow-up.

Tape limb circumferential measurements are considered an easy and practical method for monitoring the progress of lymphedema. However, several critiques have been moved against this tool for not allowing a precise assessment of limb volume. Conversely, a study showed that circumferential and CT measurements are highly complementary in the assessment of volume in the lymphedematous limb [29].

### 2.2. Water Displacement

Water displacement offers perhaps the most precise tool for the assessment of the limb volume; however, this method is impractical in clinical setting and thus seldom used. In this procedure, the patients immerse the lymphedematous limb in a container full of water. The water overflow is transferred in another box, then it is weighed and measured. Disadvantage of this method include: (1) hygienic concerns, (2) it does not provide information about swelling location, (3) is contraindicated in patients with open wounds. It is thus rarely used in clinical practice.

### 2.3. Perometry

Perometry uses an infrared optoelectronic device that can measure the volume of the swollen limb and then compared to the healthy limb. The perometer works using infrared scanning to calculate the circumference of multiple areas of the limb [30] creating a 3-D image of the limb, with the limb volume calculated in ml. A great advantage of the perometer is its capacity: (1) to measure bilateral lymphedema, (2) to localize swelling, and (3) to detect a 3% limb volume change [31].

### 2.4. Bioimpedence Spectroscopy

Bioimpedence spectroscopy (BIS) calculates the rate of electrical current transmission through the tissues by comparing impedance and resistance in the extracellular fluid between the lymphedematous limb and the healthy limb using a low-level current (<30 kHz) [32]. Advantages of this method are: (1) it is safe, painless and rapid, (2) provides objective data even for the early detection of lymphedema and (3) it is repeatable. BIS uses the impedance ratio values between the lymphedema and the healthy limb, with the latter acting as a control, to calculate the Lymphedema Index (L-Dex) ratio. L-Dex outside the range (−10 to +10) reveals early signs of lymphedema. L-Dex value increases of +10 units from baseline also support the diagnosis of lymphedema. A disadvantage is that BIS is not useful for assessing bilateral limb lymphedema.

### 2.5. Three-Dimensional Laser Scanning

Recently, three-dimensional laser scanning has been used as a promising method for the measurement of upper limb volume [33,34]. This tool allows real-time reconstruction of 3D upper limb images. Three-dimensional laser scanners showed similar accuracy and reproducibility compared to water displacement for the measurement of arm volume [33,34]. Indeed the technique shows higher intra-rater reliability compared to water displacement. Furthermore, three-dimensional laser scanners are able to identify very small differences of limb volume, including increases or reductions of swelling as a consequence of CDT [35]. Conversely, the high costs of the devices, difficulties in the detection of upper limb reference points and time-consuming nature for the elaboration of data are the main issues of this tool. A recent study showed the reproducibility and reliability of three-dimensional laser scanner compared to tape circumferential measurements to assess arm volume in BCRL patients before and after CDT pointing out the easy learning curve of this method [36].

### 2.6. Lymphoscintigraphy

Lymphoscintigraphy is currently the ‘gold standard’ imaging technique for the diagnosis of extremity lymphedema when the clinical diagnosis is uncertain and, indeed, provides a clear image of the lymphatic drainage status of the upper limb [37,38]. Lymphoscintigraphy involves injection of a radiotracer in the hand and analysis of proximal lymph node uptake. It is, generally, performed as a qualitative analysis to evaluate the following features: (1) presence or absence of axillary/elbow lymph node uptake; (2) presence of linear, dilated or absent lymphatic ducts; (3) presence and location of dermal backflow. Some centers have reported also quantitative analysis based on decay-adjusted uptake and lymphatic transport index; however these are not commonly performed [39,40]. Recently, single photon emission computed tomography-computed tomography (SPECT-CT) lymphoscintigraphy has been used for the diagnosis of lymphedema providing 3-D live images of lymph flow [38,41,42]. A recent study reported significant association between the type of dermal backflow, the lymph flow pathways, and the visualization of lymph nodes around the clavicle [42].

### 2.7. Computed Tomography (CT)

This imaging study is able to differentiate between lymphedema, cellulitis, and generalized edema [43]. CT can detect lymphedema features including skin thickening, honeycombing or presence of fat lobules. It provides a standardized and reproducible method to measure the limb volume providing a 3-D representation of the lymphedematous limb [29].

### 2.8. Indocyanine Green (ICG) Lymphography

Nowadays, indocyanine green (ICG) lymphography is the most used imaging modality for the assessment of the severity and treatment in extremity lymphedema. This imaging technique involves the intradermal injection in the distal limb of the fluorescent dye ICG. Using a near-infrared camera, a laser light source is able to show the fluorescence in the dye when functioning lymphatics are present. Instead, non-functioning lymphatics will not be visualized. Several advantages have been described for ICG lymphography such as: (1) less invasiveness without radiation and (2) the capacity to clearly observe superficial lymphatic channels in real time bedside or even intraoperatively [44]. However, the main drawback of this imaging technique is its inability to visualize deep lymphatic at more than 1 cm in depth.

### 2.9. Magnetic Resonance Lymphangiography

Magnetic resonance (MR) lymphangiography is a safe imaging technique, with high spatial resolution with the possibility to provide visualization of the function of the lymphatics. Additional MR lymphangiography features include: (1) the amount of fat deposition, (2) the muscle compartments and (3) precise limb volume [45].

## 3. Severity of BCRL and Patient Selection

Since the severity of lymphedema starts from a soft pitting edema to an irreversible non-pitting edema with fatty and fibrotic deposition, it is imperative to understand the different lymphedema stages. A number of classifications and staging systems, based on clinical and imaging findings have been proposed in the medical literature. These classification systems are further explained in Table 2.

### 3.1. International Society of Lymphology (ISL) Classification

The International Society of Lymphology (ISL) classification is the most widely used one and divides the severity of lymphedema into three stages [46]. Briefly, patients are classified as Stage 0 (latent or sub-clinical lymphedema) when lymphatic channels have been injured with impaired lymph transport, but swelling or edema is not measurable. Stage I (spontaneously reversible lymphedema) is considered with measurable swelling and pitting of the skin due to accumulation of lymph, which decreases with limb elevation or compression garments. Stage II (spontaneously irreversible lymphedema) occurs when significant adipose tissue deposition and protein-rich fluid accumulation prevent limb elevation alone or compression garments from being an effective method to reduce symptoms. In late Stage II, the limb may present increase of fat and fibrosis. Finally, Stage III (lymphostatic elephantiasis) is the most severe stage of lymphedema. It is characterized by severe swelling, excess deposition of fat and fibrosis and significant skin thickening in the form of acanthosis or hyperkeratosis.

Campisi et al. have published a similar classification with Stage I described as initial or irregular edema, Stage II defined as persistent lymphedema, Stage III as persistent lymphedema with lymphangitis, Stage IV as fibrolymphedema, and Stage V when elephantiasis is manifest [47].

### 3.2. NECST Classification

Mihara et al. have advocated a four-stage classification based on the pathological progression of post-mastectomy lymphedema. These stages are based on the histochemical changes of the lymphatic channels after axillary dissection. The changes in lymphatic channels were classified as normal, ectasis, contraction and sclerosis (NECST) [48].

### 3.3. Arm Dermal Backflow and MD Anderson Classifications

The Arm Dermal Backflow classification (ADB) [21,49], and the MD Anderson staging (MDA) [16] methods are widely used to define the severity of BCRL and both use ICG lymphangiography. The first was based on the examination of 20 patients, and the latter on 30 patients. Both staging systems include 6-stages of lymphedema severity, with stage 0 as normal linear lymphatics with no dermal backflow and stage 1–5 showing abnormal lymphatic patterns with various degrees of dermal backflow. Recently, Jørgensen et al., validated the two staging systems based on ICG lymphography, MDA Scale and ADB scale, in 237 unilateral BCRL [50]. They found near-perfect inter-rater and intra-rater agreement for both ICG lymphography staging and substantial agreement between the MDA and the ADB scales. Indeed, they found a slight correlation between the two ICG lymphography staging systems’ results to conventional circumferential measurements. They concluded that the two ICG lymphography staging were reliable, safe tools with the MDA scale providing better disease stratification than the ADB scale.

### 3.4. Cheng’s Lymphedema Grading and Taiwan Lymphoscintigraphy Staging

Cheng’s Lymphedema Grading is a 5-grade classification that includes objective symptoms, limb volume measurements, and functional evaluation of lymphatic system using lymphoscintigraphy [51]. The five grades are divided based on the limb circumferential difference between the two limbs, the affected and non-affected as follows: grade 0 (<9%), grade I (10–19%), grade II (20–29%), grade III (30–39%) and grade IV (>40%).

Recently, the Taiwan Lymphoscintigraphy Staging has been validated and incorporated into the Cheng’s Lymphedema Grading being it more objective and with the aim to offer a reliable and useful lymphedema staging system for diagnosis, severity and treatment of extremity lymphedema [19,20,37]. Patients selection for surgical treatment using the Cheng’s Lymphedema Grading is as follow: Patients with Cheng’s Grading 0 showing a range of circumferential difference between 0 and 10% and Taiwan Lymphoscintigraphy Stages L-0, P-1 or P-2 are suggested to be treated with compression garment treatment. Patients with Cheng’s Grade I and early Grade II presenting respectively a circumferential difference range of 11–20% and 20–30% are commonly treated with LVA when presenting Taiwan Lymphoscintigraphy Stages P1–P3 and linear lymphatic ducts at ICG lymphography. Instead, when they show Taiwan Lymphoscintigraphy Stages P-3/T-4/T-5 with dermal backflow at ICG lymphography, they are suggested to be treated with VLN transfer. Patients with Cheng’s Grade III and IV showing respectively a range of circumferential difference 30–40% and >40% with Lymphoscintigraphy Stages T4-T6, a single or double VLNT transfer is performed [52].

## 4. Treatments for BCRL

Current treatment options for BCRL include conservative and surgical treatments; however, determining the best treatment method for each patient remains challenging.

### 4.1. Conservative Treatments

CDT is widely accepted the universal first-line therapy for extremity lymphedema. It includes manual lymph drainage (MLD), skin care, specialized exercises, compression garments and self-education [6]. CDT is divided into Phase I Decongestion, and Phase II Maintenance and should be individualized to improve its effectiveness and contain costs.

Several advantages can be obtained by a CDT including: (1) reduction of lymphedema volume, pain and arm heaviness, (2) improvement of lymphatic drainage, (3) acceptable quality of life and (4) reduction of episodes of cellulitis [53,54]. Although conservative therapy alone may provide enough symptomatic relief, it depends essentially on patient compliance and their capacity to wear life-long compression garments.

#### 4.1.1. Manual Lymphatic Drainage

Manual lymphatic drainage (MLD) is a massage method increasing the transport capacity of the lymph collectors and moving lymph fluid and protein absorption when the lymphatic ducts are still functioning. A meta-analysis showed that, compared with other CDT modalities, additional MLD is unlikely to produce a proper reduction in the lymphedematus limb circumference [55]. In the other hand, another systematic review found that when MLD was used in combination with compression garments, provide increased swelling reduction in BCRL patients compared to the compression bandages alone, especially for moderate lymphedema stages [56].

#### 4.1.2. Compression Bandages and Compression Garments

Compression bandages are an important part of CDT maintaining the therapeutic effects of MLD. Compression bandages apply: (1) a resting pressure during the limb relaxed and (2) a working pressure when muscles contraction push the skin against resisting bandages. Low-stretch bandages produce the highest working pressure with multi-layered compression bandaging.

Compression garments are an essential part of CDT and with the aim to keep the volume reduction achieved with MLD and bandaging. Compression garments produce a two-way stretch in both longitudinal and transverse direction with the greatest pressure above the wrist and less pressure in the arm. The longitudinal pressure facilitates the joint movements. Generally, patients with BCRL wear a full arm sleeve and, frequently, a glove to prevent dermal backflow. There is no consensus regarding suitable compression values. Class 2 compression garments with 30–40 seamless are often recommended to be wear at least 12 h per day [19]. Of note, compression garments should be custom-made by a certified and experienced therapist in fitting garments for lymphedema patients.

#### 4.1.3. Exercises and Life-Style

Exercises are an integral part of CDT with the aim (1) to promote lymph flow, (2) to mobilize the joints, and (3) to strengthen the muscles. It is widely known that participation in exercises during and after oncological treatment can improve the physical and psychosocial condition, ameliorating the quality-of-life [57]. Recent studies reported that gradual weight-lifting program does not worsen the risk of BCRL compared to patients without exercises [58,59].

### 4.2. Surgical Treatments

Many surgical procedures to treat BCRL have been propose as follow: (1) physiologic procedures (lymphaticovenous anastomosis, vascularized lymph node transfer) and (2) excisional procedures (reduction or liposuction) (Table 3).

#### 4.2.1. Physiologic Procedures

In recent years, with the advent of microsurgical and supermicrosurgical techniques [60,61,62,63,64], lymphatic microsurgery procedures have gained popularity for the treatment of BCRL. Commonly practiced procedures include lymphovenous anastomosis (LVA) and vascularized lymph node (VLN) transfer. These surgeries try to deal with physiologic impairment resulted from cancer-related lymphedema and have the ability to provide venous shunting of lymphatic fluid bypassing areas of damaged lymphatics creating new lymphatic connections or by replacing the damaged lymph nodes and lymphatic channels [65].

##### Lymphovenous Anastomosis (LVA)

Lymphovenous anastomosis (LVA) is not a new procedure as it was initially described in 1969. It is a delicate supermicrosurgery technique, diverting lymph into the venous system bypassing proximal obstruction [66]. LVA has been shown to be especially beneficial in patients with early-stage upper limb lymphedema (Cheng’s Grade I and early II) [16]. In a prospective study of 100 LVAs, symptomatic improvement was described in 96% of BCRL patients. Other advantages of LVA include decreased episodes of cellulitis. Recently, Cheng’s group reported more effective lymph drainage in both proximal and distal sites using side-to-end LVA configuration compared with end-to-end LVA, without need of postoperative compression garment [17].

Previous studies have reported that LVA seems more effective in early-stage lymphedema due to the unavailability of functional lymphatic ducts in advanced stage lymphedema [16]. Therefore, advanced stage lymphedema was considered a relative contraindication for LVA [67]. However, recently Hong’s group showed promising results using LVA for advanced stage lymphedema [68]. The authors pointed out the crucial role of preoperative magnetic resonance lymphangiography and ultrasound for the success of the procedure.

Prophylactic LVA have been also performed and has successfully prevented upper limb lymphedema in 23 patients who underwent oncologic resection for breast cancer treatment and ALND [69,70].

Disadvantages of these procedure include (1) its technical difficulty for the execution anastomosing lymphatic ducts with a diameter of 0.5–0.8 mm with subdermal venules of 0.6–1.0 mm in diameter. (2) the requirements of supermicrosurgery instruments, high resolution microscope, and ICG lymphography (3) difficulty to monitor the anastomoses patency. Reported complications of LVA include infection (3.9%), lymphorrea (4.1%) and necessity of reintervention (10%) [71].

##### Vascularized Lymph Node (VLN) Transfer

VLN transfer is the latest physiological procedure added to the treatment repertoire and it is commonly indicated in more advanced cases of lymphedema. Several donor sites have been described of VLN transfer including groin, submental, supraclavicular nodes, thoracic, and omental. In 2006 Becker et al. popularized for the first time the procedure with the publication of groin VLN transfer for postmastectomy lymphedema [72]. After that, Cheng and colleagues described anatomic and clinical application of both groin and submental VLN transfer transferred into the distal limb [28,73]. Three recipient sites have been described for upper limb lymphedema such as axilla, elbow and wrist. The decision of recipient site is taken based on the severity of the lymphedema, recipient vessel availability, and surgeon preference.

Recent studies have shown the benefit of VLN flap with significantly improvement of lymphedema limb without patent lymphatic ducts compared to CDT or LVA [74]. Indeed, microsurgical breast reconstruction do not improve the outcome of postmastectomy lymphedema [74,75]. A meta-analysis compared the outcome of VLN transfer and LVA in extremity lymphedema [71]. The result showed that although both procedures were both efficient in a short-term outcome, patients with VLN transfer presented significant better improvement in the long-term with good likelihood of discontinue to wear compression garments.

VLN transfer is suggested for Cheng’s Grade II-IV who did not present patent lymphatic channels using ICG lymphography. Additional procedures such as flap debulking and liposuction following VLN transfer are suggested for Cheng’s Grade III and IV. In a recent study, patients with different grades of bilateral limb lymphedema underwent LVA in the less severe limb and VLN transfer in the more severe limb. This individualized treatment achieved effective improvement in the reduction of each limb swelling and cellulitis, as well improvements in quality-of-life [76]. Although VLN transfer has shown favorable results, however it could carry the risk of donor site lymphedema [25,77,78]. Other complications include flap loss, lymphocele, infection, and wound healing complications.

#### 4.2.2. Excisional Procedures

The first surgical method used to treat BCRL lymphedema was reported by Sistrunk in 1927 [79]. The excess skin and soft tissue were removed using a spindle-shaped incision in the medial region of the arm with removal of the deep fascia and creating a connection between superficial and deep lymphatics. Later, with Thompson a further step forward in the BCRL treatment was achieved using a lymphatic transposition method. A deepithelialized rectangular hinge skin flap was harvested from all length of the arm with the flap tip embedded near the neurovascular bundle with the aim to bridge the superficial and deep lymphatics [80].

Nowadays, excisional procedures, such radical reduction with preservation of perforators [81], and suction-assisted lipectomy [82] aim to eliminate the affected tissue in severe lymphedema stages. All excisional procedures produce the following advantages: (1) decrease limb size, (2) reduce episodes of cellulitis, and therefore improve the quality of life of the patients. Although these surgical procedures can be immediately effective to reduce the lymphedema volume, however they can carry some risks including wound complications, swelling recurrence, and the need for the patient to wear compression garments lifelong to prevent recurrence.

##### Liposuction

Fat accumulation is one of the pathologic findings of BCRL. Adipose tissue deposition is probably due because it is an endocrine organ in which complex structures of cytokine-activated cells, and chronic inflammation play a role [82]. However the pathophysiological mechanism of adipose tissue accumulation in lymphedema still remains controversial. Tashiro et al. reported adipose tissue alterations in extremity lymphedema using macroscopic and ultrasound findings [83]. They found in adipose tissue samples larger adipose lobules in lymphedema limb compared to non-lymphedema samples. Indeed, lymphedema samples presented hypertrophic changes of adipocytes and increased collagen fibres. Finally, adipose-derived stem cells and M2 macrophages were less in in lymphedema adipose tissue than in the healthy controls [83].

Liposuction is currently the most accepted excisional procedure. Brorson et al. showed that BCRL with nonpitting edema treated with liposuction presented 68% to 93% of fat, 32% of interstitial fluid, and 7% of lymph [84,85]. This excisional technique is able to remove fat producing significant arm reduction [84,86,87]. Indeed, a reduction in episodes of cellulitis was reported. A possible explanation of reduced cellulitis may be the increased skin blood flow after liposuction that could eliminate bacteria that entered through skin wounds [88]. However, the main drawback is the need to use life-long compression garments [84,89].

#### 4.2.3. Combined Treatments

Due to lack of consensus among the experts regarding the most appropriate protocol for lymphedema treatment, each surgeon applies a surgical procedure based on his personal approach. A combined treatment have been proposed as an alternative to the single strategy [65,90]. Recently, Di Taranto et al., reported that patients with extremity lymphedema treated with combined VLN transfer, LVA and liposuction LVAs showed better improvement in terms of circumference reduction compared to patients treated only with VLN transfer and liposuction [91].

Later Baumeister et al. described a new method for the treatment of 28 BCRL patients in which autologous lymphatic grafting is initially performed to bypass the axilla reestablishing lymphatic flow and later on liposuction is performed as a second step [92] without the need for additional treatments.

## 5. Conclusions

BCRL is a debilitating and chronic and condition that can severely affect the patient’s quality of life. An improvement in identification, prevention, and management of affected patients is imperative in reducing BCRL. A particular attention should be given to all stages of breast cancer treatment in order to reduce the incidence of BCRL. The use of new technologies for performing mastectomies and sentinel lymph node biopsy or axillary lymph node dissection could be useful [93,94,95,96]. Accurate physical examination and assessment of the lymphedema severity are essential to provide more predictable outcomes. A prompt management of the disease in a multidisciplinary team is the key to obtain good results [97,98,99,100,101,102,103,104,105]. Despite the fact lymphedema is still considered an incurable disease, in the last decade promising results with significant reduction of the limb swelling and improvement of psychosocial well-being have been shown.

## Figures and Tables

**Table 1 jpm-11-00402-t001:** Comparison between Different Diagnostic Tools for the Diagnosis of Breast Cancer-related Lymphedema.

Diagnostic Tool	Lymphedema Features	Advantages	Disadvantages
Circumferential Measurements	Circumferential difference	Easy and economic To monitor the progress of the disease	Not provide a precise volume assessment
Water displacement	water overflow	Reliable Validated	Hygienic concerns Not provide information about swelling localization Contraindicated in patients with open wounds
Perometry	Infrared scanning with calculation of multiple areas of the limb	To measure bilateral lymphedema To localize swelling To detect 3% limb volume change	Not available in all centers Expensive
Bioimpedence Spectroscopy	Impedance Ratio between the limbs. Lymphedema Index (L-Dex) ratio	Safe, painless and rapid Early detection of lymphedema Repeatable	Not appropriate for bilateral lymphedema Expensive
Three-Dimensional Laser Scanner	Real-time digital reconstruction of 3D upper limb	Able to identify extremely small variations of arm volume	High costs Difficulty in arm reference points detection and acquisition Time-consuming for software elaboration
Computed Tomography	Skin thickening Honeycombing Fat lobules	Objective method for limb volume	Radiation exposure Expensive
Lymphoscintigraphy	Axillary/Elbow LNs Lymphatic ducts Dermal backflow	Gold standard for the diagnosis Provide assessment of the lymphatic obstruction severity (partial or total) Allows assessment of deep lymph flows	No standardized protocol Occasional fuzzy images No detailed information on subdermal lymphatics
ICG Lymphography	Superficial Lymphatic ducts Dermal backflow	Detailed visualization of superficial lymphatic ducts Visualization and mark of lymphatic ducts intra-operatively No radiation exposure	Can only visualize lymphatics about 1.5 cm into the subcutaneous tissue
Magnetic Resonance Lymphangiography	Lymphatics Fat deposition Muscle compartments Precise limb volume	No radiation exposure Good information on the lymphatics function	No available in all centers Technically demanding Expensive

LNs: lymph nodes; CT: computed tomography; MR: magnetic resonance.

**Table 2 jpm-11-00402-t002:** Staging and Classification for the Severity of Breast Cancer-related Lymphedema.

Staging	Method	Staging Features	Characteristics
International Society of Lymphology (ISL)	Physical findings	0: latent/sub-clinical I: spontaneously reversible II: spontaneously irreversible III: lymphostatic elephantiasis	Widely accepted
Campisi	Physical findings	I: initial/irregular edema, II: persistent LE III: persistent LE with lymphangitis IV: fibrolymphedema V: elephantiasis	Rely primarily on physical findings
Arm Dermal Backflow	ICG lymphography	0: No dermal backflow 1: Splash pattern around the axilla 2: Stardust limited between olecranon and axilla lymphangitis 3: Stardust distal to olecranon 4: Stardust involving the hand 5: Diffuse and stardust pattern involving the entire limb	Safe Information regarding the lymphatic flow for LVA planning
MD Anderson	ICG lymphography	0: No dermal backflow 1: Many patent lymphatics and minimal dermal backflow 2: Moderate number of patent lymphatics and segmental dermal backflow 3: Few patent lymphatics with extensive dermal backflow 4: Dermal backflow involving the hand 5: ICG does not move proximally to injection site	Safe Information regarding the lymphatic flow for LVA planning
Cheng’s Lymphedema grading	Circumferential difference and lymphoscintography	0: 0–9% 1: 10–19% 2: 20–29% 3: 30–39% 4: >40%	Objective method
Taiwan Lymphoscintigraphy Staging	Lymphoscintography	L-0: Normal Lymphatic Drainage P-1, P-2, P-3: Partial Obstruction T-4, T-5, T-6: Total Obstruction	Validated, Reliable

LE: Lymphedema; ICG: Indocyanine Green (ICG) Lymphography; LVA: Lymphovenous anastomosis.

**Table 3 jpm-11-00402-t003:** Available Treatments for Patients with Breast Cancer-Related Lymphedema.

Treatment	Indication	Advantages	Disadvantages
Complex Decongestive Therapy	CLG 0-I	Reduction lymphedema volume, pain and arm heaviness Improvement lymphatic function Acceptable quality of life Reduction episodes of cellulitis	It is a purely symptomatic treatment Needs patient compliance Life-long compression garments.
Lymphovenous anastomosis	CLG I- early II	Safe Reduces of Circumference Reduces callulitis	Technically difficult Needs supermicrosurgery instruments Needs high resolution microscope Needs ICG lymphography Difficult to monitor the anastomoses patency
Vascularized Lymph Node Transfer	CLG late II-III-IV	Improvements in circumferential measurements, episodes of cellulitis, and quality of life	Requires intraoperative techniques of greater complexity Higher risk for postoperative re-exploration and the flap inset Risk of donor-site lymphedema
Liposuction	CLG III-IV	Decrease limb size Reduces episodes of cellulitis Improve quality of life	Risks of swelling recurrence Life-long compression garments

CLG: Cheng’s Lymphedema Grading.

## Data Availability

Not applicable.

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
