# Peer review of "Breast Cancer-Related Lymphedema: Recent Updates on Diagnosis, Severity and Available Treatments"

_jpm, 2021, doi:10.3390/jpm11050402_

Round 1
Reviewer 1 Report
There are not strict guidelines on BCRL treatment and this makes the topic somehow hard to discuss. The structure should be improved along with the depth of the discussion in several points.
Please provide a point-by-point reply
Overall
Extensive syntax and grammar editing should be accomplished. I suggest a careful revision of the entire manuscript to improve readability.
Diagnosis of BCRL
The truncated cone formula should be mentioned regarding “tape circumferential measurements”. Please expand the bibliography discussing the following:
Brorson H and Höijer P. Standardised measurements used to order compression garments can be used to calculate arm volumes to 426 evaluate lymphoedema treatment. J Plast Surg Hand Surg. 2012;46(6):410-415.
Three-dimensional laser scanning was recently proposed as a reliable and reproducible diagnostic mean in breast cancer related lymphedema. Please provide a paragraph about this point.
Page 5 section water displacement. There is a mistake in the numbered list.
Page 7. Nowadays, ICG is a cornerstone in Lymphedema diagnosis and treatment. I would clarify the advantages (I.e. bedside use, itra-op use) and disadvantages (deep lymphatics). Please expand the bibliography discussing the following:
Jørgensen MG, et al. Prospective Validation of Indocyanine Green Lymphangiography Staging of Breast Cancer-Related Lymphedema. Cancers (Basel). 2021 Mar 26;13(7):1540.
Treatments for BCRL
Throughout the entire section please have the treatment methods ordered by level of invasiveness (i.e. conservative, physiologic, liposuction and other excisional methods)
Section: Ablative and excisional procedures.
The authors should discuss the indications (in which stage? and if there is agreement about it in the current literature), the pros and cons of each listed method.
Please, put the focus on liposuction (currently it is the most accepted excisional procedure) discussing the followings:
Tashiro K, et al. Pathological changes of adipose tissue in secondary lymphedema. Br J Dermatol. 2016;77:158–167.
Brorson H and Svensson H. Skin blood flow of the lymphedematous arm before and after liposuction. Lymphology. 1997;30:165–172.
Please provide a reference to support the use of Charles procedure for the upper limb. Otherwise delete it.
Please discuss Thompson procedure for the upper limb through the following reference:
Thompson N. Buried dermal flap operation for chronic lymphedema of the extremities: Ten-year survey of results in 79 cases. Plast Reconstr Surg. 1970;45:541–548.
Section “Physiologic Procedures”
Page 13 there is a typo at the third line. “23”
Which are the outcomes of LVA in later stages?
Please improve discussion about combined strategies. I would suggest expanding the bibliography discussing the following recent studies
Di Taranto G, et al. A prospective study on combined lymphedema surgery: Gastroepiploic vascularized lymph nodes transfer and lymphaticovenous anastomosis followed by suction lipectomy. Microsurgery. 2021 Jan;41(1):34-43.
Baumeister RGH, et al. Microsurgical lymphatic vascular grafting and secondary liposuction: Results of combination treatment in secondary lymphedema. Lymphology. 2020;53(1):38-47.
Author Response
Response: We have now carefully revised the readability of the manuscript. We hope that you will find our changes satisfactory.
Diagnosis of BCRL
The truncated cone formula should be mentioned regarding “tape circumferential measurements”. Please expand the bibliography discussing the following:
Brorson H and Höijer P. Standardised measurements used to order compression garments can be used to calculate arm volumes to 426 evaluate lymphoedema treatment. J Plast Surg Hand Surg. 2012;46(6):410-415.
Three-dimensional laser scanning was recently proposed as a reliable and reproducible diagnostic mean in breast cancer related lymphedema. Please provide a paragraph about this point.
Page 5 section water displacement. There is a mistake in the numbered list.
Page 7. Nowadays, ICG is a cornerstone in Lymphedema diagnosis and treatment. I would clarify the advantages (I.e. bedside use, itra-op use) and disadvantages (deep lymphatics). Please expand the bibliography discussing the following:
Jørgensen MG, et al. Prospective Validation of Indocyanine Green Lymphangiography Staging of Breast Cancer-Related Lymphedema. Cancers (Basel). 2021 Mar 26;13(7):1540.
Response: Thank you for your insightful comments. We have now discussed the truncated cone formula and added the Three-dimensional laser scanning in the diagnostic tools section. Indeed, we agree that ICG lymphography needed more discussion.
Treatments for BCRL
Throughout the entire section please have the treatment methods ordered by level of invasiveness (i.e. conservative, physiologic, liposuction and other excisional methods)
Section: Ablative and excisional procedures.
The authors should discuss the indications (in which stage? and if there is agreement about it in the current literature), the pros and cons of each listed method.
Please, put the focus on liposuction (currently it is the most accepted excisional procedure) discussing the followings:
Tashiro K, et al. Pathological changes of adipose tissue in secondary lymphedema. Br J Dermatol. 2016;77:158–167.
Brorson H and Svensson H. Skin blood flow of the lymphedematous arm before and after liposuction. Lymphology. 1997;30:165–172.
Please provide a reference to support the use of Charles procedure for the upper limb. Otherwise delete it.
Please discuss Thompson procedure for the upper limb through the following reference:
Thompson N. Buried dermal flap operation for chronic lymphedema of the extremities: Ten-year survey of results in 79 cases. Plast Reconstr Surg. 1970;45:541–548.
Response: Thank you for your excellent comments. We reordered the level of invasiveness and we focus our attention on liposuction discussing the articles suggested. We also included the Thompson procedure
Section “Physiologic Procedures”
Page 13 there is a typo at the third line. “23”
Which are the outcomes of LVA in later stages?
Please improve discussion about combined strategies. I would suggest expanding the bibliography discussing the following recent studies
Di Taranto G, et al. A prospective study on combined lymphedema surgery: Gastroepiploic vascularized lymph nodes transfer and lymphaticovenous anastomosis followed by suction lipectomy. Microsurgery. 2021 Jan;41(1):34-43.
Baumeister RGH, et al. Microsurgical lymphatic vascular grafting and secondary liposuction: Results of combination treatment in secondary lymphedema. Lymphology. 2020;53(1):38-47.
Response: Thank you again. We agree that more discussion regarding LVA in advance stages and combined strategies was needed and we add thispart as a separate section.

Reviewer 2 Report
A thorough coverage of the subject with extensive literature review. English is generally good. However, it does not read as well as it could or make a coherent story. I am not sure how much it helps me treat patients
Author Response
Response: Thank you for your comment. We have carefully revised the manuscript based on the Reviewer 1 suggestion. We hope that you will find our changes satisfactory.

Round 2
Reviewer 1 Report
The authors responded to the previous comments